# Assess of Combinations of Non-Pharmacological Interventions for the Reduction of Irritability in Patients with Dementia and their Caregivers: A Cross-Over RCT

**DOI:** 10.3390/brainsci12060691

**Published:** 2022-05-26

**Authors:** Tatiana Dimitriou, John Papatriantafyllou, Anastasia Konsta, Dimitrios Kazis, Loukas Athanasiadis, Panagiotis Ioannidis, Efrosini Koutsouraki, Thomas Tegos, Magda Tsolaki

**Affiliations:** 11st Department of Neurology, Aristotle University of Thessaloniki, 44 Salaminos Street, Halandri, 15232 Athens, Greece; ekoutsou@auth.gr (E.K.); ttegos@auth.gr (T.T.); tsolakim1@gmail.com (M.T.); 23rd Age Center IASIS and 2nd Neurology Department, Attikon Hospital, 16562 Athens, Greece; jpapatriantafyllou@gmail.com; 31st Department of Psychiatry, “Papageorgiou” General Hospital of Thessaloniki, Aristotle University of Thessaloniki, 54124 Thessaloniki, Greece; konstaa@auth.gr (A.K.); loukatha@outlook.com.gr (L.A.); 43rd Neurology Department, Aristotle University of Thessaloniki, 54124 Thessaloniki, Greece; dimitrios.kazis@gmail.com; 52nd Department of Neurology, Aristotle University of Thessaloniki, 54124 Thessaloniki, Greece; ispanagi@auth.gr

**Keywords:** irritability, BPSD, dementia, randomized trial, cross-over trial, non-pharmacological

## Abstract

Introduction: Dementia is a very common disorder that affects people over 65 years old all over the world. Apart from the cognitive decline, Behavioral and Psychological Symptoms of Dementia (BPSD) are a crucial matter in dementia, because they affect up to 90% of the patients during the course of their illness. Irritability has been found to be a common BPSD and one of the most distressing behaviors for the caregivers. The aim of the current study was to explore the efficacy of a combination of non-pharmacological interventions to treat irritability. Methods: Sixty patients with different types and stages of dementia with irritability were participated in a cross-over RCT. Three non-pharmacological interventions were used; (a) Validation Therapy (VT)/Psycho-educational program, (b) Aromatherapy/massage and (c) Music Therapy (MT). The study assessed the three non-pharmacological interventions in order to find the most effective combination of the interventions. This study did not compare pharmacological and non-pharmacological treatments. The interventions lasted for five days. There was no drop-out rate. All patients were assessed at baseline using Mini Mental State of Examination (MMSE), Addenbrooke’s Cognitive Examination Revised (ACE-R), Geriatric Depression Scale (GDS), Functional Rating Scale for symptoms in dementia (FRSSD), and Neuropsychiatric Inventory (NPI) (sub questions for irritability). Only NPI used for the assessment after each intervention. The analyses used categorical variables, Wilcoxon signed-rank test, Chi-square test and z value score. Results: The most effective combination of non-pharmacological interventions was Aromatherapy/massage (*p* = 0.003)-VT plus Psycho-educational program (*p* = 0.014) plus MT (*p* = 0.018). The same combination was the most effective for the caregivers’ burden, too (*p* = 0.026). Conclusions: The above combination of non-pharmacological interventions can reduce irritability in patients with dementia and caregivers’ burden.

## 1. Introduction

General improvements in daily living conditions have radically increased life expectancy in general population. However, the prevalence of different pathologies and syndromes are related to age increase [1], because it has not been found yet a way to increase the survival time and improve the quality of life of the elderly, at the same time. Dementia is a very common disorder that affects people over 65 years old, all over the world. Behavioral and Psychological Symptoms in Dementia (BPSD) is a term that includes several behaviors of patients with dementia (PwD). BPSD represent a heterogeneous group of unwanted behaviors, such as psychotic, affective, and behavioral symptoms that occur in the majority of PwD and they cause severe problems to the patients and caregivers’ distress [2]. According to the Neuropsychiatric Inventory (NPI) the most common behaviors are; delusions, hallucinations, agitation, depression, anxiety, euphoria, apathy, disinhibition, irritability, wandering, sleep disorders and appetite problems [3]. BPSD play a major role in the prognosis of dementia, and can lead to early institutionalization, cognitive decline, functional impairment, problems in daily activities, reduced independence, and increase caregivers’ distress [4]. In addition, BPSD are a crucial matter because they affect approximately the 60–90% of PwD [3]. The etiopathogenesis of BPSD is complex. Probably they are a result of the interaction of multiple factors, such as biological (brain changes), psychological (personality) and social-environmental factors (daily living) [5].

Irritability is a common BPSD [6]. It has been found to be one of the most distressing behaviors to caregivers, as well [4]. Irritability is one of the precursors for anger and aggressive behavior, however it is a discrete behavior. Studies have shown that irritability exists in most common types of dementia and contributes to caregivers’ burden [7].

The pathogenesis of BPSD has not been clearly described, however it seems to be a result of a complex interplay of cognitive, social, biological, and psychological parameters [8]. It seems that high scores of irritability on the NPI are associated with lower fractional anisotropy of the anterior cingulate in PwD (specifically in Alzheimer’s disease (AD) patients and patients in the Mild Cognitive Impairment (MCI) stage) [9]. Additionally, when irritability is accompanied with affective symptoms (such as depression), or psychosis, there are no clear evidence for specific neurobiological substrates. A FDG-PET study in AD patients has demonstrated that irritability has metabolic changes in the bilateral middle, posterior cingulate gyri and right temporal and right frontal [6]. Moreover, internal and external environments contribute to irritability. In some cases, irritability may be also related to a pre or/and co-existing psychotic disorder. Irritability may be worsening when the patient is hungry, sleepy or in pain [10,11].

The principle of Validation Therapy (VT) is the acceptance of the reality of the PwD. VT uses some behavioral and psycho-therapeutic techniques; non-threatening words, speaking in a clear and calm tone of voice, rephrasing unclear communication and respond gentle to non-verbal communication [12]. 

Aromatherapy in combination with massage therapy (AM) has shown promising results on the reduction of some BPSD (aggressive behaviors, anxiety and sleeping disturbances) [13]. Nevertheless, the mechanisms of action of aromatherapy are yet unknown [14], however lavender oil, melissa- based and lemon balm oil have been reported with antioxidant actions of vitamin E, which improves the state of blood vessels close to the skin. This is why there are the most common used oils [15,16].

Finally, MT can be classified as either receptive (listening to music) or participatory (making music). Previous studies have shown beneficial effect of MT in different types of BPSD [17,18]. It is a pleasurable activity with no side-effects. 

The current pharmacological treatment for the irritability is: cholinesterase inhibitors, memantine, antidepressants, antipsychotics and benzodiazepines. However, antipsychotics affect cognitive decline and are not recommended for more than 12 weeks [19]. Antipsychotics may increase risk of stroke and death [20]. First generation or typical antipsychotics show lack of tolerability and have many side effects, such as sedation, anticholinergic effects, and extrapyramidal symptoms. Second generation or atypical antipsychotics have higher tolerability, however high attention should be paid to their side effects, as well, such as high cholesterol, diabetes, seizures, drowsiness, weight gain, extrapyramidal symptoms, hypotension, hyperprolactinemia, and insomnia [2,21,22]^.^ Benzodiazepines, on the other hand, should be avoided in PwD, because they are associated with cognitive decline [23]. In addition, medical cannabidiol 3% is suggested as a potential treatment for the BPSD in general [24], and irritability in PwD. However, the mechanism of cannabidiol is not well elucidated, and there is much clinical evidence of its effectiveness [25]. Furthermore, the cholinesterase inhibitors have shown some positive results in the treatment of irritability, however their side effects should be well considered [26,27]. Their side effects include: diarrhoea, stomach cramps, increased production of saliva and excessive sweating [28]. Therefore, there is a need for effective non-pharmacological interventions. 

The aim of the current study is to explore the efficacy of a combination of methods to treat dementia with non-pharmacological interventions. 

## 2. Materials and Methods

### 2.1. Subjects

In this study, we included sixty patients with dementia and irritability symptoms from the Neurological Departments of the General Hospitals of Thessaloniki and Athens. The patients were diagnosed in accordance with the ethical principles (declaration of Helsinki). The sample was diagnosed with different stages and types of dementia; the Alzheimer’s Disease (AD), Vascular Dementia (VAD), Lewy Body Dementia (LBD), Dementia in Parkinson’s disease (PDD), Frontotemporal Dementia (FTD), Mixed type (AD & VAD) and AIDS. The participants have been informed and their caregivers have given consent. There was no dropout rate. The sample was randomly assigned in six (6) different groups of ten (10) participants each. In that way the randomization of the study was remained. Table 1 shows the baseline characteristics of the sample. Twenty-seven (27) participants were males (45%). The average age of the sample was 73.52 years old (SD 8.4) and the average years of education was 10.1 (SD 4.81). The scores were not corrected by education.

### 2.2. Procedure

This is a cross-over randomized controlled trial. The study assessed the non-pharmacological interventions in order to find the most effective combination of treatments. Because the effective control of some BPSD is hard, and the literature so far has not support effective non-pharmacological interventions for some unwanted behaviors, such as irritability, the goal was to combine some effective, pleasurable, and non-harmful interventions, in order to reduce the behavior and caregivers’ burden, as well. For that reason, the NPI questions and sub-questions for irritability was applied to the family caregivers at the beginning of the process. The results were recorded and then the patients were randomly assigned into 6 different groups of 10 participants each. Every group received the same non-pharmacological interventions, but on a different sequence. The sequence of the interventions among groups is shown on Table 2. Each treatment was taken place to Monday- Friday and Saturday was used for the assessment. 

### 2.3. Interventions

The interventions were chosen based on four factors: (a) they should be easily performed by the unprofessional caregivers, (b) they were some evidence in the literature that the interventions have some benefits on the reduction of irritability, (c) they are pleasurable and (d) they have no known side-effects.

Some effective non-pharmacological interventions for the reduction of other BPSD have been used in the current study, in order to evaluate their effectiveness.

In order for the caregivers to be able to perform VT, a psycho-educational program (based on ASPAD project) was conducted. The program started 3 months before the trial begun. It was administrated to all family caregivers either in face to face or online meetings. The duration of the psycho-education program was 2 weeks and included 24 seminars. Every seminar lasted approximately 2 h. The seminars were referring to general knowledge about dementia, its progress, BPSD, non-pharmacological interventions and daily challenges. One private personal counseling session (60 min) was also included. The instruction in the “Psycho-educational therapy” was “to do nothing” (ignore) when irritability appears. 

The Aromatherapy/massage was administrated in the back and lower limbs for 20 min. every morning after breakfast. The current study used lavender. 

In the current study we used the preferable music of each patient. The intervention applied for 45 min.per session, once per day, every morning after breakfast. The length of the intervention seems to be in accordance with previous studies [29]. 

### 2.4. Measures

Mini Mental State Examination (MMSE) [30,31]: MMSE is a 30-point questionnaire that is used to evaluate the cognitive status. It is used to estimate the severity of cognitive decline. The questionnaire examines registration, attention, recall, language, and orientation. Higher scores indicate better cognitive performance and lower scores severe cognitive decline.

Addenbrooke’s Cognitive Examination Revised (ACE-R) [32,33]: ACE-R is a 100-point questionnaire that is used to evaluate the cognitive impairment. It includes MMSE. It is highly sensitive and can be used for the diagnosis of dementia. It includes questions about orientation, registration, attention, concentration, recall, verbal fluency, memory, language, spatial abilities, perceptual abilities, and recognition. Higher scores indicate better cognitive performance.

Geriatric Scale of Depression (GDS) [34,35]: This scale is a questionnaire of 30 questions that examines if the patient has depression. The patient answers with a YES/NO. Higher score indicates higher level of depression.

Functional Rating Scale for Symptoms in Dementia (FRSSD) [36,37]: It is a scale to access the Activities of Daily Living. The scale is a questionnaire to the caregiver and includes 14 different daily activities, such as: eating, dressing, incontinent, speaking, sleeping, faces’ recognition, personal hygiene, name memory, fact memory, alertness, agitation, space orientation, emotional status, socializing. The scale is scored from 0–3 (whereas 0 = fully independence and 3 = fully dependence). 

Neuropsychiatric Inventory (NPI) [3,38]: The questionnaire is administrated to the caregiver. The questionnaire evaluates the frequency and severity of the symptom as long as the impact that each behavior has on the caregiver. Frequency is scored from 0–4 (0 = rarely happens, 4 = happens every day), severity from 1–3 (1 = mild severity, 3 = severe) and the distress is scored from 0–5 (0 = not at all, 5 = extremely). The domain total score is the product of: (a) frequency X severity score and (b) the total score of caregivers’ distress. A total score is obtained by summing all the domain total scores. The questions of NPI for irritability are:−Does the resident have a bad temper, flying “off the handle” easily over little things?−Does the resident rapidly change moods from one to another, being fine one minute and angry the next?−Does the resident have sudden flashes of anger?−Is the resident impatient, having trouble coping with delays or waiting for planned activities or other things?−Is the resident easily irritated?−Is the resident argue or is he/she difficult to get along with?−Does the resident show any other signs of irritability?

### 2.5. Data Analysis

Categorical variables were presented as percentages while continuous variables were presented as Mean value and Standard Deviation (SD). Wilcoxon signed-rank test used, because the distribution of the differences between the samples cannot be assumed to be normally distributed. Chi-square test was used in order to find differences in gender in the 6 groups and finally z value score was used in order to find the type of dementia in each group. *p* values less than 0.05 were considered statistically significant. SPSS 25.0 (IBM Inc., Armonk, NY, USA) was used for the statistical analysis.

## 3. Results

The Mean scores of all the patients were; MMSE 20.52 (SD 4.81), ACE-R 56.67 (SD 18.95), GDS 8.57 (SD 6.08), FRSSD 15.32 (SD 12.69), NPI Result 7.82 (SD 2.39), NPI Distress 3.43 (SD 0.78). According to the diagnosis, the 55% of the patients suffer from AD, the 18.3% from MCI, the 1.7% from AIDS, the 6.7% from VAD, the 1.7% from LBD, the 11.7% from PDD, the 3.3% from FTD and the 1.7% from Mixed dementia (Table 3). The most effective combination for the reduction of irritability was: Aromatherapy/massage, followed by VT/psychoeducational program, followed by MT. The same combination was the most effective for the reduction of caregivers’ burden, as well. Specifically, group 4 applied the interventions with the above-mentioned sequence and reduced baseline NPI (8± 2.75). Aromatherapy/massage when applied first reduced the behavior (*p* = 0.003). VT/psychoeducational program when applied after aromatherapy/massage reduced irritability (*p* = 0.014) and MT when applied after VT/psychoeducational program reduced irritability further (*p* = 0.018). In group 4 of the caregivers the same combination reduced the behavior statistically significant. Specifically, the baseline NPI (4 ± 1.07) was reduced when aromatherapy/massage applied first (*p* = 0.026), followed by VT/psychoeducational program (*p* = 0.032), followed by MT (*p* = 0.035). The results of all groups are shown on Table 4 and Table 5.

## 4. Discussion

As we recorded previously, to our knowledge, there is not any study about combinations of non-pharmacological interventions for the treatment of irritation or other BPSD. Only one old review confirmed the positive results of massage therapy in combination with MT in agitated patients [39]. For that reason, we are going to discuss about the results of the non-pharmacological interventions that we used, as they were used in the past, separately. 

The literature so far lacks large sample sizes and suffers from several limitations, such as: compliance with the interventions, subjective judgments and therefore unclear results and medication usage at the same time with the non-pharmacological intervention that might interfere with results [40].

A recent review mentions that aromatherapy is based on the use of plant products or oils and can be delivered through massage [12]. This systematic review found positive effects of aromatherapy and massage in the management of irritability and agitated behaviors in general. Another large recent review (AMSTAR = 8) confirms the positive results of the previous systematic review [41]. This recent review included 7 RCTs, although only two had usable data. According to these two trials, the first one reported positive result [42], whereas the second one did not mention any significant difference [43]. In the first trial 71 participants completed the trial [30]. The sample was randomly assigned to aromatherapy with Melissa essential oil (N = 36) or placebo (N = 36). The oil was applied to patients’ faces and arms twice a day. The intervention lasted for 4 weeks. The second trial [31] that did not find significant differences was a double-blind parallel-group placebo-controlled randomized trial that conducted in England. One hundred fourteen participants completed the study. However, the authors mention that there is no evidence that aromatherapy is superior to placebo or donepezil [31]. The review underlines some limitations, such as: the trials used different scales, limited sample size and problems in methodological quality [29]. Another review with AMSTAR = 6 [44] identified 11 trials reports promising results of “aromatherapy and massage” however, it is crucial to mention that this review included only one RCT [45]. Furthermore, in terms of massage therapy the literature has shown beneficial effects, as well. Some large reviews underline that massage therapy can have positive results on the reduction of anxiety, depression, and agitated behaviors in dementia [11,46,47]. It is critical to mention two other RCTs with beneficial results of the aromatherapy and massage. Specifically, the first trial is a cross-over randomized study, that compared the experimental group with the placebo group (used lavender) [48]. The other one used acupoint that was pressed for 2 min.with lavender oil for no longer than 15 min. once per day for 5 days a week for 4 weeks. The results found improvements on the general agitated behaviors of the PwD [49]. Moreover, there is no reliable dosage, and the duration of the massage is not yet confirmed. Likewise, the type of oil that is most effective is yet unknown [50]. In terms of massage therapy, it can be applied to different parts of the body (back, arms, hands, shoulders, legs, neck etc.) with positive results [51]. Yet, there is a strong need for further research, because despite the above-mentioned limitations, aromatherapy and massage is a safe intervention, inexpensive and enjoyable for both the PwD and the caregivers, as well. 

VT on the other hand, has some studies that have shown promising results in decreasing BPSD in general [52,53] and others that pointed no significant differences [54,55,56]. There are two studies that have analyzed the benefit of VT claim positive effects of the intervention. Specifically, the first study [57] had a sample of 30 patients and the VT was given twice a week for 12 weeks. The intervention lasted 45–60 min. The intervention showed beneficial effects in contrast with the control group. The second trial was a case-control study and had a sample of 50 patients and it lasted 16 weeks. The intervention took place both during individual sessions (3 times a week for 20 min. per session) and group sessions (once a week for 45–50 min per session) [39]. VT is an inexpensive non-pharmacological intervention and has no side-effects. Nevertheless, some negative effects may appear on the caregivers because distress could be exacerbated if the caregivers are not well prepared, and they do not know exactly what to do. Additionally, the psycho-educational program that was used in the current study aimed to diminish this difficulty. The results of the previous studies are in accordance with the results of the current study, as well.

MT is a non-pharmacological intervention with promising effects in general. It is an enjoyable intervention and in some BPSD seems to be highly effective (depression, anxiety, apathy) [58]. However, the literature so far lacks much evidence that MT has an impact on irritability. It is important to refer though, that MT has some promising studies on the reduction of the agitated behaviors in PwD [59]. However, it is crucial to mention one study that found significant results of MT on irritability [60]. This study had a sample of 59 participants which were randomly assigned. The experimental group (N = 30) received 30 MT sessions for 16 weeks and the control group (N = 29) received the usual educational support or other entertaining activities. Here, arouses a matter of fact that in NPI questionnaire there are differences between agitated/aggressive behavior and irritability, but the literature so far may include symptoms of irritability in the general umbrella of agitated/aggressive behaviors. It is a matter of fact that sometimes it may be hard to separate these two behaviors. In our study, we found that MT when combined with aromatherapy/massage, and VT/psychoeducational program, may have beneficial effects on the reduction of the irritability, unless it is performed after the other two interventions.

Another finding of our study was that the same combination that can reduce irritability, can reduce caregivers’ distress, too. An explanation is that caregivers’ burden is influenced by the behavioral symptoms of the patient. Therefore, if the patient feels better or reduces the unwanted behavior, the caregiver feels less anxiety and depression, as well. Additionally, the combination of aromatherapy/massage and psychoeducational program it seems that can give us statistically significant results on the reduction of irritability. It is a combination that needs further research. It is important to mention that the current study found a combination of non-pharmacological interventions that can also reduce the caregivers’ burden, as well. It is highly important not to forget about the caregivers and their distress, when studying about the BPSD. In terms, of the most effective combination that was found in group 4, the study seems to be in accordance with previous studies, which mention that aromatherapy/massage is an effective alternative for the reduction of irritability in PwD. The intervention following by a cognitive intervention (VT/psychoeducational program) may give to the caregivers the necessary information, in order to manage irritability in a more efficient way. MT is a pleasurable intervention, as many other previous studies have also mentioned, and it seems that it helps to relax further the patients who suffer from irritability.

Regarding the strengths of the study, it is important to mention that it was a cross-over randomized controlled study. In selecting this design, the risk of bias has been diminished. The sequence of the procedure does not interfere with the results. Moreover, the trial included patients with different types of dementia, however it is understood that this may be a limitation, at the same time, as well. Apart of the heterogeneity of the sample it seems that our results have an effect to both genders, types, and stages of dementia. NPI questionnaire is a valid and reliable tool in order to identify irritability in PwD. The study was conducted with systematic attention and the psycho-educational program was strict. The clinician gave specific and clear guidance to the caregivers and interfered when needed. However, limitations do exist in the current trial. The main limitation is that the sample includes PwD of several degrees of severity. The caregivers provided the interventions by themselves. The duration of the intervention was only 5 days. This happened because the caregivers always ask for rapid solutions. A follow-up study should be required. Future researchers should focus on large sample size and randomized controlled trials in order to come to safe conclusions. As dementia is a complex syndrome a combination of non-pharmacological intervention should be also examined. 

## 5. Conclusions

In conclusion, it seems that there is a lack of evidence for the non-pharmacological approaches for irritability in dementia. The literature so far lacks large and blinded trials with methodological quality and clear results. It is important to find solutions in order to reduce irritability as it is one of the most common and disturbing behaviors in dementia. It is also crucial to help the caregivers to reduce their distress, by providing them with effective non-pharmacological solutions. The current study found a promising combination of non-pharmacological interventions that can reduce irritability and caregiver’s distress. Finally, the caregivers have to find different solutions and different combinations of non-pharmacological interventions for irritability. It is important for the caregivers to remember that gaining knowledge for expert health professionals on the subject of dementia and its difficulties can help them find better ways to help their patients and improve their quality of life, as well.

## Figures and Tables

**Table 1 brainsci-12-00691-t001:** Baseline characteristics of the sample.

	Mean (SD) or N (%)
Males, N(%)	45% (N = 27)
Age	71.4 (9.40)
Years of education	8.9 (4.21)
MMSE	17.75 (4.59)
ACE-R	53.7 (19.26)
GDS	7.33 (4.89)
FRSSD	17.22 (8.49)
NPI Results	7.15 (1.48)
NPI Distress	3.72 (0.92)

**Table 2 brainsci-12-00691-t002:** The sequence of the procedure (A = Validation Therapy (VT)/Psycho-educational program, B = Aromatherapy and Massage therapy, C = Music Therapy).

Group	Sequence	1st Week	2nd Week	3rd Week
1	ABC	A	B	C
2	ACB	A	C	B
3	BAC	B	A	C
4	BCA	B	C	A
5	CAB	C	A	B
6	CBA	C	B	A

**Table 3 brainsci-12-00691-t003:** Percentages of the different types of dementia of the sample.

AD	VAD	LBD	PDD	FTD	Mixed	MCI	AIDS
55%	6.7%	1.7%	11.7%	3.3%	1.7%	18.3%	1.7%

Abbreviations: Alzheimer’s disease (AD), Vascular Dementia (VAD), Lewy Body Dementia (LBD), Parkinson’s dementia (PDD), Frontotemporal Dementia (FTD), Mixed Dementia (AD & VAD), Mild Cognitive Impairment (MCI).

**Table 4 brainsci-12-00691-t004:** Results (patients).

Group 1	ΝΡΙ Original	ΝΡΙ before A	A–Β	Β–C
Mean score ± SD	8 ± 2.67	8 ± 2.67–7 ± 2.64	7 ± 2.64–6 ± 1.81	6 ± 1.81–8 ± 2.62
Percentiles		5.50–9, 5.50–9	5.50–9, 4–6.50	4–6.50, 5.50–9
*p*		0.370	0.039	0.780
Group 2	ΝΡΙ Original	ΝΡΙ Before A	A–C	C–Β
Mean score ± SD	8 ± 2.06	8 ± 2.06–8 ± 2.06	8 ± 2.06–8 ± 2.06	8 ± 2.06–8 ± 1.68
Percentiles		7.50–9.75, 7.50–9.75	7.50–9.75, 7.50–9.75	7.50–9.75, 5.50–8
*p*		1	1	0.963
Group 3	ΝΡΙ Original	ΝΡΙ Before Β	Β–A	A–C
Mean score ± SD	8 ± 2.41	8 ± 2.41–6 ± 1.94	6 ± 1.94–8 ± 2.17	8 ± 2.17–8 ± 2.18
Percentiles		7.50–9.75, 4–8	4–8, 6–8.25	6–8.25, 6–9
*p*		0.027	0.600	1
Group 4	ΝΡΙ Original	ΝΡΙ Before Β	Β–C	C–A
Mean score ± SD	8 ± 2.75	8 ± 2.75–6 ± 2.16	6 ± 2.16–5 ± 2.75	5 ± 2.75–4 ± 2.39
Percentiles		6.75–9, 3.75–8	3.75–8, 4–6	4–6, 2–4
*p*		0.003	0.014	0.018
Group 5	ΝΡΙ Original	ΝΡΙ Before C	C–A	A–Β
Mean score ± SD	6 ± 2.42	6 ± 2.42–6 ± 2.42	6 ± 2.42–6 ± 2.42	6 ± 2.42–6 ± 2.40
Percentiles		5.50–8.25, 5.50–8.25	5.50–8.25, 5.50–8.25	5.50–8.25, 5.50–8.25
*p*		1	1	1
Group 6	ΝΡΙ Original	ΝΡΙ Before C	C–Β	Β–A
Mean score ± SD	8 ± 2.26	8 ± 2.26–8 ± 1.77	8 ± 1.77–6 ± 1.57	6 ± 1.57–8 ± 2.26
Percentiles		6–9, 6–9	6–9, 6–8.25	6–8.25, 6–9
*p*		0.317	0.046	0.766

**Table 5 brainsci-12-00691-t005:** Results of the caregivers.

Group 1	ΝΡΙ Original	ΝΡΙ before A	A–Β	Β–C
Mean score ± SD	3.5 ± 1.05	3.5 ± 1.05–3.5 ± 0.91	3.5 ± 0.91–2.5 ± 0.82	2.5 ± 0.82–3 ± 1.03
Percentiles		2–4, 2–4	2–4, 2–3.25	2–3.25, 2–4
*p*		1	0.059	0.102
Group 2	ΝΡΙ Original	ΝΡΙ Before A	A–C	Γ–Β
Mean score ± SD	3 ± 0.70	3 ± 0.70–3 ± 0.84	3 ± 0.84–3 ± 0.67	3 ± 0.67–3 ± 0.84
Percentiles		3–4, 3–4	3–4, 3–3.25	3–3.25–1.75–3
*p*		0.836	0.564	0.559
Group 3	ΝΡΙ Original	ΝΡΙ Before Β	Β–A	A–C
Mean score ± SD	4 ± 0.51	4 ± 0.51–3 ± 0.84	3 ± 0.84–4 ± 0.70	4 ± 0.70–4 ± 0.51
Percentiles		3–4, 2–3	2–3, 3–4	3–4, 3–4
*p*		0.015	0.300	0.717
Group 4	ΝΡΙ Original	ΝΡΙ Before Β	Β–C	C–A
Mean score ± SD	4 ± 1.07	4 ± 1.07–2.5 ± 0.96	2.5 ± 0.96–2 ± 1.07	2 ± 1.07–1.5 ± 1.03
Percentiles		2.75–4, 1.75–3	1.75–3, 1–3	1–3, 1–2
*p*		0.026	0.032	0.035
Group 5	ΝΡΙ Original	ΝΡΙ Before C	C–A	A–Β
Mean score ± SD	3 ± 0.67	3 ± 0.67–3 ± 0.63	3 ± 0.63–3 ± 0.63	3 ± 0.63–2.5 ± 0.52
Percentiles		3–4, 3–4	3–4, 3–4	3–4, 2–3
*p*		0.317	1	0.038
Group 6	ΝΡΙ Original	ΝΡΙ Before C	C–Β	Β–A
Mean score ± SD	3 ± 0.70	3 ± 0.70–3 ± 0.69	3 ± 0.69–2 ± 0.94	2 ± 0.94–3 ± 0.70
Percentiles		3–4, 3–4	3–4, 1.75–3	1.75–3, 3–4
*p*		1	0.031	0.224

## Data Availability

The study did not report any data.

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
