# Peer review of "Assess of Combinations of Non-Pharmacological Interventions for the Reduction of Irritability in Patients with Dementia and their Caregivers: A Cross-Over RCT"

_brainsci, 2022, doi:10.3390/brainsci12060691_

Round 1
Reviewer 1 Report
The present work entitled ‘Non-pharmacological interventions for irritability/ liability in patients with dementia’ assessed the effects of three non-pharmacological interventions on the irritability of people with dementia and the burden of their caregivers. This BPSD has a high prevalence in most dementias, is considered a trigger of other BPSD and increases the caregiver burden. Therefore the results are of interest both for the current knowledge about these interventions and for their practical implications for a best practice and management of BPSD. However, the text seems not focused enough. It needs clarification on the goals, description of the interventions, the rationale of why studying the sequence instead of comparison, etc. Limitations are not clearly identified or defined. Other issues that need improvement are indicated below in a constructive manner.
Title
The title should be informative about the findings. As it stands right now it could be a review on non-pharmacological interventions. It does not mention the key findings: the ‘good’ combination, and the effect also in caregivers.
Abstract
The abstract needs to be rewritten. The aims of the work are not defined in the abstract.
Please, note that the authors referred to dementia as affecting people over ‘’<65 ‘’years old, with the mathematical sign <. It should be better to omit (people over 65) it or to use +65.
‘’Sixty (60)’’ is redundant. Please, use only ‘Sixty’ in the abstract and methods. Similarly, with other numerical indicators used in the text.
“Three non-pharmacological interventions used” is missing the past tense ‘were’.
“The measurements which were used are…” could be better be expressed as ‘Effects of the interventions on irritability were assessed using…
“…is Aromatherapy/massage…” please, use past tense: “… was Aromatherapy/massage…”
Introduction
BPSD are used in the text as singular or plural. Please, check and correct accordingly.
The introduction should provide a description of the interventions and supporting references, since are the topic of interest. All the references to them are offered in the methods, where there only the protocols should be detailed, so half of the description provided could move to the introduction without major problems.
The rationale to study the ABC combination instead of comparing the benefits of each one should be provided.
Materials and methods
“The sample was diagnosed with different stages and types of dementia; AD,” The ‘Alzheimer’s Disease’ is missing.
Verbs are used in present and past (i.e. years of education ‘’is’’ instead of ‘’was’’). Please, correct.
Since caregivers are also subjects of study, the sociodemographic description of caregivers should be given. That is sex, age and years of education.
In the description of the temporal procedure “”Each treatment was taken place for five days, there was two days wash-out period and at the morning of the 6th day NPI questionnaire (only irritability questions and sub-questions) were applied again, in order to record the results.””” it can be inferred that refers to Monday-Friday for interventions, Saturday for assessment. If this is like this, please, indicate, as the week distribution is as important or could be more important in terms of ‘feasibility’ for its implementation than the exact number of days. The authors also refer in the discussion that the 'five days' schedule was motivated by the demands of caregivers.
Table 2. the sequence—Table 2. The sequence
Since the years of education (8.9) have a standard deviation of 4.21, the authors should indicate in the methods if the scores were or not corrected by education.
Results
The main limitation is that the sample includes PwD of several degrees of severity, despite the NPI scores original were the key starting point.
Discussion
The first page of the discussion starts with a short indication of the findings “An effective combination of non-pharmacological interventions that can reduce irritability and caregivers burden was found. Aromatherapy/ massage has shown significant results on the reduction of irritability in previous studies, as well’ “ but is used to discuss previous studies, not the current one. The discussion should be reorganized so that the present results are discussed and supported with other works. Again, what is lacking is the discussion on why a specific sequence ABC is more effective than the other 5.
The other parts of the discussion address the issues of: dosage, type of oil, sample size and other limitations. The second finding, that refers to the benefits to reduce caregiver burden is important.
The authors refer to the “the trial included patients with different types of dementia.” as a strenght but it could also be seen as a limitation, since in each group of 10 patients several dementia types and severity degrees are included, so it is not possible to know if a specific type of dementia or stage is improved or remains similar to basal levels.
Author Response
Dear reviewer,
many thanks for your comments. They were all highly appreciated and helpful to us.
The title has been changed. The abstract was re-written and the aim of the study has been added. We used the past tense and we made all the other corrections that you have suggested. We wrote again the introduction, and we explained the rationale of the ABC combination. We add the limitations that you suggested, and the discussion is re-organized.

Reviewer 2 Report
This is a very good paper dealing with non-pharmacological strategies to treat dementia, particularly in those individuals presenting irritability and other behavioral or psychological symptoms. The paper is well written and of interest for the readers; however, several minor changes should be made befdore considering it for publication.
The subsections of the abstract are not adequately followed up according to the journal style. The abstract should be divided into introduction, methods, results and conclusions.
The authors have not compared a combination of pharmacological and non.pharmacological strategies. This should be mentioned in the abstract.
A brief statistical analyses section should be included in the abstract.
The introduction section is really brief. I would recommend to expand the first part related to the several syndromes or domains that can appear in patients with dementia. More references about pharmacological treatment of dementia are recommended in the last part of the introduction section.
The subsection about aims and objectives should be separated from the last paragraph of the introduction.
The main aim of the paper cannot be "to find a combination". The objective seems to be to explore the efficacy of a combination of methods to treat dementia with non-pharmacological tools.
How were the patients randomized? It should be explained in the methods section.
Table 3 presents the percentages of different types of dementias. Abbreviations should be explained before or after the table.
Why are results from tables in red?
In Table 4, the content of columns should be lines, and viceversa.
The discussion section is starting with concluding that an effective combination of non-pharmacological interventions were found. I recommend to be more cautelous. I think that it would be better that one combination of non-pharmacological therapies were found to be effective to treat dementia.
a strenghts and limitations section in the discussion is welcome.
I consider that a conclusions section is expected.
Author Response
Dear reviewer,
thank you for your valuable comments. We are happy to share with you, that we have divided the abstract according to the journal's style. We have added the phrase that we do not compare pharmacological to non-pharmacological interventions. We have added a brief statistics in the abstract. We have also expanded the introduction with more information and references. We have separated the aims from the objectives, and the objectives have been changed. To answer your question, some results are in red, because they refer to the most effective combinations. The red colour is to give emphasis to which group was the most effective. We have explained the abbreviations. We have added a strength and limitation section and finally we also added a conclusion section.

Round 2
Reviewer 1 Report
The authors have done the suggested amendements to improve the quality of their report and the Ms current version provides a clear and specific title on the research done that also than enhances the chances to be found and used, specific aims, more detailed methods and rationales, so all these changes give more value to the understanding of tools/strategies used and would encourage to use the 'succesful' combination under the lens of science, which in non-pharmological interventions is always difficult to achieve. The work will be very helpful for those confronting these scenarios and the need to have research data when making decisions on best strategies to be used. Therefore I'd like to sincerely THANK the team for their efforst to solve this issue.